# Implementer and recipient perspectives of community-wide mass drug administration for soil-transmitted helminths in Kwale County, Kenya

Hugo Legge[1]*, Stella Kepha[1,2,3], Mateo Prochazka[1], Katherine Halliday[1], Rachel Pullan[1], Marie-Claire Gwayi-Chore[4‡], Doris Njomo[3‡]

1 Department of Disease Control, Faculty of Infectious and Tropical Diseases, London School of Hygiene & Tropical Medicine, London, United Kingdom, 2 Pwani University Bioscience Research Centre, Pwani University, Kilifi, Kenya, 3 Eastern and Southern Africa Centre of International Parasite Control, Kenya Medical Research Institute, Nairobi, Kenya, 4 Department of Global Health, University of Washington, School of Public Health, Seattle, Washington, United States of America

‡ These authors are joint senior authors on this work.
* hugo.legge@lshtm.ac.uk

**Data Availability Statement:** Participants did not provide consent to have their full audio transcripts made publicly available. Furthermore participants

## Abstract

Soil-transmitted helminthiases (STH) are one of 17 neglected tropical diseases (NTDs) earmarked for control or elimination by 2020 in the WHO's Roadmap on NTDs. Deworming programs for STH have thus far been focused on treating pre-school and school-aged children; however, there is a growing consensus that to achieve elimination of STH transmission, programs must also target adults, potentially through community-wide mass drug administration (MDA). There is currently a gap in the literature on what components are required to deliver community-wide MDA for STH in order to achieve high intervention reach and uptake. Nested within the TUMIKIA Project, a cluster randomized trial in Kenya evaluating the effectiveness of school-based deworming versus community-wide MDA, we collected qualitative data from program implementers and recipients in eight clusters where community-wide MDA was delivered. Data collection included semi-structured in-depth interviews (n = 72) and focus group discussions (n = 32). A conceptual framework for drug distribution was constructed to help build an analysis codebook. Case memos were developed for each top-level theme. Community-wide MDA for STH was perceived as a complex intervention with key administrative and social mobilization domains. Key actionable themes included: (1) developing an efficient strategy to allocate reasonable workload for implementers to cover all targeted households; (2) maximizing community drug distributors' motivation through promoting belief in the effectiveness of the intervention and providing sufficient financial incentives; (3) developing effective capacity building strategies for implementers; and (4) implementing a context-adapted community engagement strategy that leverages existing community structures and takes into consideration past community experiences of MDAs. Transitioning from STH control to elimination goals requires significant planning and action to ensure community-wide MDA is delivered with sufficient reach and uptake. We

could potentially be identified through full release of the data. Due to this, complete transcripts will not be uploaded to a public repository. Instead, anonymized excerpts of the data will be made available on request to researchers with clear study objectives. Data requests should be made to Abigail Gross, the LSHTM LASER group Coordinator, at abigail.gross@lshtm.ac.uk who will ensure access to relevant excerpts of the data.

**Funding:** The study is supported by the Bill & Melinda Gates Foundation (#OPP1033751), with additional funding provided by the Children's Investment Fund Foundation (CIFF) and the European & Developing Countries Clinical Trials Partnership (EDCTP). SK is supported by THRiVE-2, a DELTAS Africa grant # DEL-15-011 from Wellcome Trust grant # 107742/Z/15/Z and the UK government. The funders had no role in the design of this study and will not have any role during its execution, analyses, interpretation of the data, or decision to submit results.

**Competing interests:** The authors have declared that no competing interests exist.

present findings that can inform national deworming programs to increase intervention delivery capacity.

## Author summary

Infections by soil-transmitted worms are common in tropical and subtropical areas. Control strategies usually involve distributing deworming drugs to children, who are most affected. However, recent evidence suggests that distributing these drugs to whole communities, including adults, might help to achieve the elimination of these worms as a public health problem. As part of a large trial in Kenya evaluating the distribution of deworming drugs to communities, we conducted a qualitative study to explore the perspectives of drug delivery among drug distributors and recipient communities. We conducted and analyzed in-depth interviews and focus group discussions. For community deworming to be implemented well, it requires distributors to be highly motivated and follow a plan that fits well with the characteristics of the target community. In order to accept the deworming drugs, communities need to know about the health problem and the intervention, and trust the delivery system. These findings should prove useful to national deworming programs planning to distribute drugs to whole communities when developing their delivery strategies.

## Introduction

Current control strategies for soil transmitted helminthiases (STH) are primarily focused on reducing morbidity amongst preschool-age children (PSAC) and school-aged children (SAC) using age-targeted preventive chemotherapy [1]. In many settings however, school-based deworming is insufficient to control infection at the community-level, since transmission may be maintained by infected adults. Recent evidence suggests that using a community-wide mass drug administration (MDA) strategy might substantially reduce transmission, potentially eliminating STH as a public health problem [2,3]. Given the momentum for elimination of STH in the global health community, there is a need to identify the most effective and sustainable strategy to deliver preventative chemotherapy (PC), with a potential shift from age-targeted to community-wide MDA [4].

Community-wide MDA is a key intervention for the control and elimination of several NTDs, including lymphatic filariasis (LF), onchocerciasis, trachoma, and schistosomiasis. Multiple studies have focused on the effectiveness and acceptability of MDA for these diseases [5–8]; however, there is limited evidence supporting the use of community-wide MDA for STH. The TUMIKIA Project, conducted in Kenya, was the first cluster randomized control trial assessing the effectiveness, equity, and cost of school-based versus community-wide treatment strategies on the prevalence and intensity of STH infection [9,10]. The impact results concluded that community-wide MDA was more effective in reducing hookworm prevalence and intensity across all age groups than school-based delivery, providing much needed evidence on the feasibility and effectiveness of community-wide MDA for STH control [9].

Given this emerging scientific evidence and policy environment supporting more comprehensive efforts to control STH, national NTD control programs aiming to shift from school-based to community-wide delivery platforms will need guidance on how best to implement and monitor these interventions. Moreover, understanding perspectives of key actors involved

in MDA implementation is necessary to inform programmatic improvements and help meet treatment coverage targets. Within the TUMIKIA Project, we conducted a qualitative study to explore perspectives of implementers and recipients of community-wide MDA for STH on various implementation aspects. In this paper, we present the results of an analysis focused on understanding key components of the reach and uptake of the intervention.

## Methods

### Ethics statement

Ethical approval was obtained from the Kenya Medical Research Institute (KEMRI) Ethics Review Committee (SSC No. 2826) and the London School of Hygiene & Tropical Medicine (LSHTM) Ethics Committee (7177). TUMIKIA is registered with ClinicalTrials.gov NCT02397772. An information sheet was provided to all individuals invited to participate in the study in Swahili. Participants underwent written informed consent and agreed to have IDIs or FGDs audio-recorded. Illiterate participants or participants who did not speak Swahili had the consent sheet read to them in their preferred language by or in the presence of an independent, literate witness. The participant then either signed or provided a thumb print, observed by the witness who also provided their signature. During transcription, participant names were replaced with alphanumeric unique identifiers to ensure anonymity and confidentiality.

### Study setting

The TUMIKIA Project was a community cluster randomized trial evaluating school-based and community-wide treatment platforms for STH, conducted in Kwale County, Kenya between 2015 and 2017. Full details of the trial design are published elsewhere [9–11]. In brief, standard school-based MDA using albendazole was delivered in the control arm while in the intervention arms, community-wide MDA with albendazole was delivered annually and bi-annually via door-to-door distribution over an eight-day period. The distribution period of the community-wide MDA was decided upon in collaboration with the Ministry of Health (MoH) and based on the timeline of previous MDAs in the study setting. Delivery leveraged MoH community health service structures by recruiting existing frontline health workers, known as Community health volunteers (CHVs), as drug distributors. CHV selection was overseen by the County Public Health Office and organized by the Community Health Assistants (CHAs), according to standardized guidelines for selection, which included community participation in nomination of individuals. In all but a few exceptions CHVs were selected from the same community in which they lived and as a result were treating in their own village. CHVs were required to cover approximately 120 households across eight days, and were paid using a performance-based strategy (500KSH per day–approximately USD$5.00 –consistent with nationally-recognized daily subsistence allowance rates). Terms of employment for CHVs were developed in consultation with the MoH and local government. Before each implementation round, a short training cascade was delivered, which provided general information on STH, community mobilization, planning and implementation of MDA, and completion of treatment registers. This was delivered through the sub-county to (CHAs), who managed the intervention delivery in assigned clusters. CHAs were then responsible for training and subsequently supervising CHVs, who were responsible for delivering community-wide MDA door-to-door. Community mobilization and engagement was conducted through forums involving chiefs, ward administrators and CHAs, and meetings and door-to-door visits conducted by village elders.

Qualitative data for this study were collected from implementers and recipients from the bi-annual community-wide treatment arm of the project in August 2016, one year after initiation of the trial. For the purpose of this study, "recipients" include community members, including opinion leaders within TUMIKIA intervention clusters. "Implementers" include CHAs and CHVs. Within the Kenyan health system, CHAs and CHVs are key stakeholders who support the delivery of community-based health promotion and prevention activities [12].

## Participant sampling

A total of eight villages were selected across the 80 intervention clusters (equivalent to MoH community units comprising approximately 1000 households each). These were purposively sampled on the basis of treatment uptake, sub-county and rural/peri-urban location. Within each of the four sub-counties, two clusters were stratified as high or low coverage based on the results of coverage surveys from the previous rounds of MDA. A village was selected from each of the eight resulting strata, with an aim to select a range of peri-urban, rural and remote communities. Within selected villages, CHVs, opinion leaders, and community members were purposefully selected according to availability and participation in the MDA. Selection of opinion leaders aimed to cut across various influential groups in the community, including religious leaders (Muslim and Christian), teachers, chiefs, village elders, businessmen and political leaders. Semi-structured in-depth interviews (IDIs) were held with community opinion leaders, CHAs and CHVs. In addition, focus group discussions (FGDs) were conducted with community members; up to 12 participants were invited to each FGD. Groups were structured based on age (18–34, >34) and sex. Eligibility criteria included age ≥18 years and willingness to participate in the study. Participants completed a brief demographic questionnaire and received a 300KSH (approximately USD$3.00) transport compensation.

## Data collection

The IDIs and FGDs were conducted within three months of the third round of MDA by a facilitator in the preferred language of the participants, including Swahili, Duruma and Digo. Interview and discussion guides were developed based on literature reviews and adapted from past studies conducted on community-wide MDA [13]. Each FGD was conducted by two facilitators, one of whom asked questions while the other took notes. Both IDIs and FGDs were audio recorded and held in private areas to ensure participants' confidentiality. Audio recordings were transcribed verbatim in the interview language and then translated into English by trained field officers.

## Data management and analysis

A conceptual framework mapping out the key components of community-wide MDA was developed as a tool to shape the codebook and coding process. Development of the framework was based on existing literature and authors' implementation experience regarding delivery of MDA programs [14–17]. The framework (Fig 1) presents key inputs and contextual factors relevant to the planning and delivery of MDA leading to two outputs: "reach" and "uptake". We define reach as the ability of implementers to make contact with target individuals and offer them treatment. Uptake, often referred to as compliance, is defined as the successful receipt of the tablet by target individuals (i.e. swallowing the tablet) once offered.

Coding was conducted using NVivo 12.0. A preliminary cycle of coding allowed for the codebook to be refined with additional codes added or removed as analysts (HL and SK) became familiar with the data. Once this cycle was complete, the codebook was finalized and used to complete coding of all study transcripts. A subset of transcripts was coded by all assigned coders and was assessed to ensure that high inter-coder reliability was being

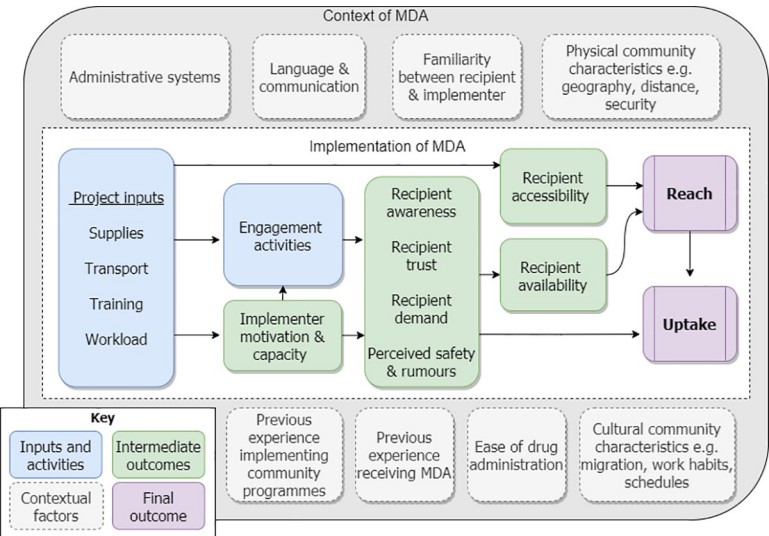

**Fig 1. Conceptual framework for delivery of community-wide mass drug administration.**

maintained (k>0.65). The process was repeated until the level of agreement reached was acceptable. After all transcripts were coded, data aggregation queries were used to develop analytic case memos using Microsoft Word. Written case memos were stratified by respondent type and structured according to main themes and sub-themes within the conceptual framework. While our sampling approach included selecting villages from high and low coverage clusters to ensure maximum variability, we did not compare themes by coverage since the focus of the study was on understanding factors related to successful MDA implementation, regardless of reported coverage. Throughout the process, analysts met regularly to discuss codebook updates, case memos and thematic saturation. Results of this analysis are presented in a narrative summary of key themes, a conceptual framework, and discussed as actionable recommendations. Representative quotes were embedded within the results to exemplify themes, with minor grammatical alterations to improve readability.

## Results

A total of 411 respondents were included in this study (Table 1). Results are organized in two sections: reach of households within the target population, and uptake of deworming drugs by community members.

**Table 1. Types of interviews conducted and number of participants per interview.**

| Interview | Type | Total number (*N = 411*) |
|---|---|---|
| *FGD** | Young women (18–34 years) | 77 (k = 8) |
| | Young men (18–34 years) | 87 (k = 8) |
| | Adult women (>34 years) | 91 (k = 9) |
| | Adult men (>34 years) | 84 (k = 7) |
| *IDI* | Opinion leaders | 32 |
| | Community health assistants (supervisors) | 8 |
| | Community health volunteers (drug distributors) | 32 |

* "k" represents number of FGDs conducted per respondent type. Each FGD group included 8–12 individuals.

### Reach

Successful delivery of community-wide MDA was seen by respondents to be dependent on adequate planning of logistical and administrative aspects of the intervention. Key attributes to achieve high reach included: (1) a well-trained and motivated workforce of community drug distributors; (2) a flexible and responsive implementation strategy; and (3) understanding and addressing factors that shape community availability.

### "Let them be trained and be recognized": Implementer training and motivation

Implementers described factors that influenced their motivation to participate in the MDA and their intention to participate in future MDAs. Primary among these was their belief in the effectiveness of the intervention, which provided them with the opportunity to improve the health of their community, their sense of self-efficacy in carrying out their roles, and the level of financial and non-financial incentives, such as knowledge and recognition offered to them by participating in the intervention:

> *"The reason that [I will] join the drug distribution next time is that I want my people to get the cure, because if I left this to those [CHVs] who are new in this area [community members] will miss the drugs." (CHV)*

> *"I thought [the MDA] was good because good health is needed by everyone . . . good health brings happiness. Another reason is moving in the community helps the community know you." (CHV)*

> *"You also add us something. The last four days we were being given only [mobile phone credit]. Add us at least lunch allowance . . .The thing [is] my CHVs have not been trained. So humbly let them be trained and be recognized." (CHA)*

Implementers mentioned technically and culturally tailored training that is responsive to their needs as being a critical component for successful MDA delivery. They agreed on the need for training focused on developing their capacity to either manage CHVs or to be successful distributors themselves. However, they noted that these trainings should be non-repetitive, especially as implementers gain expertise with each subsequent round of MDA. For training sessions to be successful, implementers noted that they should provide relevant and updated information for their specific roles during MDA planning and delivery, be well-timed (i.e. not conflicting with other MoH activities or significant cultural events), and have sufficient time to cover all topics.

> *"Ok generally since we have been in TUMIKIA for one and a half years, they have to focus on key areas mostly to concentrate on forms, because the issue of understanding the helminths is now clear, so they should concentrate on forms when there any changes. Because the last time we did deworming the forms were different from the ones we used, so you get problems." (CHA)*

### "Either they increase the number of days or they increase the CHVs": adaptable implementation strategy and efficient supply chain

Given the complexity of implementing community-wide MDA and the perceived short time-line for drug administration (8 days), respondents often mentioned efficiency and equity of

workload as key factors for consideration while developing the MDA implementation strategy. For many CHAs and CHVs, workload assignment was a major priority, including the number of days they were provided to administer deworming medicines and the size of the team that was assigned to plan, deliver, and monitor treatment. Across all locations, workload was commonly referenced as being excessive for implementers who noted that the magnitude of the geographical area and target population being covered affected their work efficiency:

> *"The time [for distribution] was short . . .the households were not many but they were far apart. . . It's possible to just do the return in one day for the person [going to a village] which is near. . ., but for someone like me who come from far, [it] can be a problem." (CHV)*

A common suggestion was to increase the number of CHVs and/or the number of days for treatment to cover the allotted number of houses, especially if a CHV was assigned large villages or ones that were far from their own homes:

> *"I can say these days are not enough. . . Every time they ask why these villages are not always covered I told them I was doing it alone and it's a big village, so it's either they increase the number of days or they increase the CHVs" (CHV)*

Implementers also emphasized the importance of ensuring that key materials were provided in adequate amounts at each point of the supply chain, especially drugs and treatment forms. CHVs emphasized the need for travel distance between storage points to be minimized, highlighting that long distances from storage points to target communities resulted in increased time spent travelling or greater expenditure on transport.

> *"[By] the time I have finished the drugs I am very far from where I collect the drugs. I had to walk or use [a] motorbike to come for the drugs." (CHV)*

For CHAs, in addition to geographical barriers, they noted that their areas of supervision should be contiguous and the number of clusters assigned to one CHA should be minimized for ease of providing operational and technical support to their CHVs. Additionally, implementers noted the importance of considering other ongoing community health priorities by MoH and scheduling treatment days to take place around religious or cultural events (e.g. Ramadan).

As an additional dimension to workload, implementer respondents also highlighted how interactions with household members were often time-consuming due to education efforts taking place within households, including time set aside to respond to concerns before household members accepted treatment. Respondents linked these interactions with longer working hours which in turn had the potential to impact overall reach. Despite the intense workload, CHVs worked long hours to ensure that they reached all households.

> *"You leave in the morning and come back at night because all that process of reaching a household–you introduce yourself, explain what you have gone to do and then you convince them until they agree. It takes a long time." (CHV)*

Several implementers perceived community-wide engagement prior to MDA as the best method to reduce the need for "doorstep" education, further increasing their efficiency and improving reach:

*"What I think is, before the distribution just like the way the drugs are being distributed the information should be passed the same way.. this makes the distributors work easy as they will save time explaining." (CHV)*

### "You won't find them": community availability

Respondents perceived the availability of community members as a key determinant of reach. Community availability was hindered by competing responsibilities, including school, casual labor, or employment that led members to be absent from the household:

*"If only one person and the guide were to distribute to all houses maybe they arrive at around noon when most women go to the shamba (garden) and the children have gone to play so it was difficult." (Community member, youth male FGD)*

*"You can revisit even three times but you won't find them. Because in our area if a man is not employed he goes to [the] quarry and mine stones." (CHV)*

Migration of entire households was also cited as representing a challenge for CHVs as it sometimes required multiple trips to confirm whether a household was truly vacant or simply absent at the time of the visit.

*"Whenever I went [to the household] he was not there, until the last day when the neighbor told me that I was tiring myself and that he was in Mombasa." (CHV)*

More broadly, revisiting households due to unavailability of community members (CHVs were instructed not to leave tablets for unavailable household members, and instead conduct call-backs at a later date) was time-consuming for implementers, especially when returning to more isolated households as CHVs were forced to dedicate time and resources to trace relatively few individuals.

Implementers, for the most part, did not perceive engagement activities to be effective in promoting community availability as they could only relay a wide range of days over which drug administration would take place. However, community respondents suggested that if it was possible for community engagement messages to include more precise information on when drug delivery would occur, this could potentially promote community availability.

*"The way [to deliver community engagement activities] which could have been used is that people could have been called for a meeting and then be told . . .we have called you to make you aware that there will be deworming drug distribution on this date and those who will bring the drugs will be in company of [the] chairman. That could have made people aware and stay at their homes that mentioned day and date waiting for the drugs." (Opinion leader)*

### Uptake

Trust and demand were perceived to be key and consistent determinants of MDA uptake. Community members referred to these to validate their own or others' decisions whether to accept the deworming drugs, or storing or disposing them after distribution. Both trust and demand emerged as multidimensional and interrelated themes, involving: (1) perceived safety, rumors and beliefs about the deworming medicines; (2) perceived effectiveness and need for deworming and risk of STH infection; and (3) community awareness and knowledge of the

intervention. Community engagement activities involving agents of influence in the community were important to ensure uptake given their ability to build trust and demand.

## "Medicine, or other things?": perceived safety, rumors and beliefs

Perceiving deworming drugs as safe was key for MDA uptake. Individual perceptions on safety among community members were shaped by past experiences taking these drugs or those from other MDA campaigns. These perceptions were based on specific experiences of others in the community, including children or neighbors, who had experienced side effects in the past. Under these circumstances, taking the medication was framed as an act of trust, overcoming the fears associated with perceived side effects:

> "They will [be] interested in taking [the drugs] because even those who did not take, they are asking themselves why our fellows used the drugs and they were not affected. So there are some who will gain courage to use the drugs." (Community member, youth male FGD)

Additionally, fearful perceptions and beliefs about MDA were fueled by rumors circulated in the community. Respondents mentioned that their communities believed MDA did not intend to control STH, but that it was a strategy for family planning, or for another harmful purpose. These beliefs affected trust and demand for the intervention:

> "Some swallowed the tablets some didn't. Those who did not some said they heard rumors that the drugs are meant for family planning and they can cause strokes and elephantiasis" (Community member, youth female FGD)

> "When one is given the drugs they just take the drug to end the conversation [with CHVs] and then later they throw it, because the drug may be illuminati wanting to take their blood." (Community member, youth male FGD)

Even in the presence of information that directly addressed rumors, some community members refused to take the drugs. This behavior appeared to stem from lack of familiarity with CHVs and deep mistrust and resistance to the intervention:

> "Anything related to health, the villager believes on the doctor, like the drugs. But when I take them as me, [the villagers] will have doubts, has this person brought me medicine, or other things?" (CHV)

> "They may be distributed by my enemy and he/she may poison me or add some other drugs on it." (Community member, youth male FGD)

Knowing the CHVs as members of the community beforehand also influenced trust, as did their formal appearance, which affected the perceived legitimacy of the intervention:

> "If they [CHVs] are all strangers there will be resistance." (Community member, adult male FGD)

> "The distributors should have good appearance and look attractive because some distributors approach you when very rough, even if you don't have the appetite of taking the drugs." (Community member, youth male FGD)

Implementers considered formal uniforms such as branded project t-shirts and bags to be of particular value when drug distributors were not familiar to their assigned community:

*"T-shirt[s] were not [available] so we just distributed [the drugs and] because we are known to the community people believed us. But if it were in the village that we were not known it would have [had] effects." (CHV)*

## "Maybe I am being treated for an illness I don't have!": perceived effectiveness and need

Despite the rumors and fears, most recipients appeared to perceive MDA as effective for treating helminth infections, based on their own or others' previous experiences. Effectiveness was occasionally linked to the perceived resolution of symptoms like diarrhea, cramping, poor appetite, abdominal distention, and the visible passage of worms with stools. Some participants mentioned worm infections were now less common in the community, and that the need to seek care for worm infections among children was now rare. These perceptions of effectiveness consolidated demand for the intervention:

*"I think it was fine because many people have less problems since we had children with worms and after taking the drugs and slept at night, the worms were getting out and now the problems have ceased. So the drugs have helped people a lot." (Opinion Leader)*

The demand for community-wide MDA targeting STH was also justified by the perceived environmental risk of infection given poor water and sanitation infrastructure:

*"In our village the toilet is only one, so without giving the drugs for worms the entire village will have worms." (Opinion Leader)*

In contrast, a few respondents questioned their need for taking medications if they were asymptomatic, or did not have an infection confirmed with diagnostics, perceiving distributing drugs to the whole community as an ineffective strategy:

*"But the drugs were given to everybody, how did they know each and every one has worms? The process was not perfect at all. Maybe I am being treated for an illness I don't have!" (Community member, youth male FGD)*

The recurrent nature of MDA campaigns also affected the perceived effectiveness of deworming medicines, particularly in communities that previously received MDA for other diseases, challenging both trust and demand. Community members were rarely able to discern between the focus of current and previous campaigns for other infections.

*"Some said these drugs are not for worms and that they are brought frequently, if they are for worms they were brought before, so why are they being brought again? So it's like some had doubt." (Opinion Leader)*

## "Reduce the speculations": community engagement activities, community awareness and knowledge

When discussing drivers for mistrust and lack of demand for treatment, respondents suggested poor community awareness and knowledge about the MDA, since this could facilitate rumors, false beliefs, and affect perceptions of need. Implementers and recipients alike perceived community engagement activities as acceptable and effective strategies to improve uptake via awareness, knowledge and trust-building. Respondents highlighted that community

engagement activities were reliant on the participation of community authorities who activated a cascade of social mobilization across pre-existing community leadership structures, including village chiefs, sub-chiefs, and elders. Limited involvement of any of these groups could jeopardize the dissemination of engagement messages, and thus negatively impact MDA uptake:

> *"We should be informed through public meetings by chairman and this will lead people to know the importance because if it happens [that] a CHV comes to my home without informing me, no way I will consume the drugs."* (Community member, youth male FGD)

Because these stakeholders are influential figures that inspire respect and authority, their involvement was key to build trust:

> *"They [the drugs] were left behind for me, and when I came back my wife gave them back to me. I asked who had left them and I was told, it's [CVH name]. I remembered that the chairman had told me about it, I didn't hesitate or throw them away."* (Community member, youth male FGD)

> *"When the day reaches and the distributor reaches your house you will remember, oh, there's a day that the chief said there are drugs that will be distributed."* (Community member, youth male FGD)

Sufficient spacing between community engagement activities and MDA was required to ensure messages were sufficiently processed and circulated in the community. Preferred channels for community engagement included posters, verbal messages from personnel at health facilities, mass messaging through radio, information delivered to children during school-based MDA, and door-to-door messaging involving CHVs and community leaders. Simultaneous use of multiple channels was favored, particularly those that involved culturally-appropriate verbal messages, since illiteracy was reported as a frequent problem in the communities.

During drug distribution, community members expressed expectations of CHVs to act as gatekeepers of knowledge and to provide information otherwise unavailable to them. The distribution of drugs without face-to-face interaction with a CHV was seen as insufficient to motivate uptake. This perception was especially prevalent if CHVs left drugs at the household for future consumption amongst absent household members:

> *"My father, for instance, he wasn't at home and the drugs were left for him, we tried to explain to him but he didn't take [them], also I wouldn't take [them] in the same case."* (Community member, youth male FGD)

In addition, CHVs selected to be drug distributors were perceived by community members to be important and intrinsically trustworthy, which in turn improved uptake.

> *"Since the CHV bears the name 'daktari wa nyanjani' (doctor) . . .they [the community] could just accept to be given the drugs by the CHVs automatically."* (CHA)

In contrast, perceiving CHVs as misinformed, improperly trained or unable to provide sufficient information led to mistrust and refusal to take the deworming medicines:

> *"In my village, those who did not take the drugs are there but later they used them, they refused because the [CHV] did not have knowledge to explain to them clearly on the*

*importance and use the drugs so that they can understand better.*" (Community member, youth male FGD)

While community engagement activities might adequately address perceived safety, efficacy and lack of knowledge, deep mistrust seemed to be a systemic problem:

"*Mostly here in our area ignorance is common, we get the message but we despise the message . . .*" (Community member, youth male FGD)

## Discussion

This study explored the perspectives of implementers and recipients on the delivery of a community-wide MDA program for STH in Kwale County, Kenya. In this context, community-wide MDA for STH was characterized as a complex intervention with multiple, interacting administrative and social components. Reach of MDA depended on effective training and sufficient motivation of drug distributors, an efficient supply chain, an adaptable and achievable implementation strategy, and an adaptive response to community availability. Uptake was driven by trust in the intervention and its implementers as well as demand for treatment. Locally adapted community engagement activities that leveraged community structures to increase community awareness and knowledge about the intervention were key, particularly when tailored to address local perceptions on effectiveness and safety shaped by previous MDAs. Our findings highlight key strategies (Table 2) that are instrumental for effective MDA delivery and are relevant to inform planning of future community-wide MDA programs that aim to control or eliminate STH.

**Table 2. Themes associated with reach and uptake and actionable mitigation strategies.**

| Theme | Impact on reach | Impact on uptake | Mitigation strategies |
|---|---|---|---|
| Development of an efficient and reasonable workload | Workload impacts the efficiency of administering treatment, specifically (i) the travel to and interaction with individual households, and (ii) the number of days and human resources allocated for MDA delivery | Effective interaction between drug distributors and household members facilitates trust<br>Sufficient time for callbacks increases likelihood of directly observed treatment | Ensure the sufficiency of: number of drug distributors and supervisors and their allocated households, the time for MDA, provision of transport<br>Plan distribution strategies so as to maximize the availability of target communities<br>Develop adaptable implementation strategies that respond to feedback from/ needs of implementers and recipients |
| Motivated and appropriately selected drug distributors | Effective recruitment and incentives sustain motivation and performance | CHVs promote trust with recipients when they are (i) already familiar to the community in which they are distributing, and (ii) identifiable through a uniform | Develop an effective recruitment strategy and accompany incentive structure (both intrinsic and financial/material)<br>Use of drug distributors that live and work in the area that they are from so to engender trust |
| Developing effective capacity building strategies for implementers | Training facilitates ability to plan, deliver, and monitor MDA | Well-trained knowledgeable CHVs engender trust with recipients | Develop capacity building activities that avoid repetition of material, have sufficient time, consider varying degrees of experience, and are scheduled to avoid overlap with ongoing responsibilities |
| Design of a contextually-relevant community engagement strategy that leverages existing community structures | An engaged community has less need for last-minute, doorstep education, thus increasing CHV availability during delivery | Strong community engagement strategies impact levels of trust in the intervention and demand for the treatment | Consider community experiences with past MDAs, address common rumors and fears surrounding MDA<br>Use existing community structures, ensure sufficient duration and spacing of messages, & utilize multiple communications channels |

First, the **design of an efficient and reasonable workload strategy** is key to both reach and uptake. When implementers reported a heavy workload assignment, they directly linked this to lower reach, as less time was available to contact all households on their treatment list and conduct any necessary call-backs. This is supported by findings from previous studies assessing MDA campaigns for LF in Ghana and Kenya, where distributors specifically highlighted that insufficient MDA duration was linked to lower reach [18,19]. Moreover, uptake was also impacted by overly ambitious treatment quotas as time-pressured CHVs opted to leave tablets at households for absent individuals rather than return to ensure directly observed treatment, which previous studies have linked to increased drug uptake [20]. In order to ensure that workload assignment is efficiently calibrated, implementers must consider the time requirements for all constituent aspects of drug delivery. This includes the number of households allocated to a drug distributor, the time required to collect the drugs and travel between households, which depends on access barriers including geography, weather and transportation, the time spent at each household to educate and treat members, and the additional time needed to conduct call-backs. In addition to this, the findings point towards the need for active communication channels to be in place between program planners, implementers, and participants throughout the intervention that allow for stakeholder feedback to result in adaptations in MDA planning and delivery. If any of these factors are not taken into consideration during the planning phase of an MDA, reach and uptake could be negatively impacted.

**Appropriately recruited and motivated drug distributors** were another important element of effective drug delivery. Motivation to participate and continue to participate as a drug distributor was linked by respondents to belief in the effectiveness of the intervention, the provision of financial incentives, and their anticipation of social recognition. To retain experience in the program and allow recruitment of high quality volunteers, programs should ensure that drug distributors are trained, so as to understand the need for treatment, and that they are financially incentivized to the point where the associated costs of volunteering are covered as well as the opportunity cost of participation. These results agree with findings from a review of qualitative studies of LF MDAs that highlighted the importance of financial and altruistic incentives for implementer retention [17]. The identity of drug distributors was also highly important as the recruitment of distributors from among the local community was key to the promotion of trust between recipients and implementers, as has been previously shown by studies on LF MDA in Kenya and Tanzania [13,21]. Further to this, the provision of official identification materials such as a uniform or branded bags can be used to promote uptake through inferring legitimacy on the drug distributor. This replicates outcomes reported in a review of factors impacting MDA delivery for NTD programs, which stated that the acceptability of the distributor among community members was increased when they were presented wearing officially branded materials [22]. These findings piece together a set of perspectives held by respondents that suggest that while aspects of drug delivery should be professionalized (for example payment of drug distributors and formalized work attire) maintaining some approaches from the community directed treatment (ComDT) model (such as drug distributors selected from within the communities in which they are working) could be beneficial to the delivery of MDA. This is relevant to the PC-NTD community as a whole as some MDA programs move away from the ComDT approach and towards the use of paid health workers as drug distributors [23]. These results suggest that there are important aspects of community participation that should not be abandoned when this transition takes place.

**Locally adapted and appropriately scheduled training** for implementers was seen as being vital to their ability to carry out the basic function of their roles beyond drug administration, such as completing treatment logs and compiling coverage data. Most importantly, training was necessary in order to provide drug distributors with the requisite knowledge to explain

the need and benefits of MDA to community members and address any concerns. Insufficiently trained drug distributors were likely to jeopardize uptake as their lack of knowledge endangered recipients' trust in the intervention. These findings are corroborated by studies on MDA for LF and onchocerciasis in Tanzania, where poor training was linked by respondents to limited ability to allay community members' concerns on drug safety [21,24]. With the exception of ensuring sufficient time is given towards training, previous studies have highlighted the importance of training without providing further detail on what factors contribute towards effective training [17,21]. In addition to ensuring sufficient training time, our findings suggest that care must be given to ensure that trainings avoid excessive repetition of well understood themes, such as basic health information, and instead focus on new elements of the MDA such as any changes to drug delivery or M&E processes. In addition, trainers should be aware of the varying levels of experience present amongst drug distributors and tailor their trainings to ensure that all needs are catered for.

Our study also highlights the need for **contextually-relevant community engagement strategies,** which appear to be vital to uptake by promoting trust and demand. The importance of community engagement activities has been widely referenced in previous studies on MDAs for a number of diseases including malaria, schistosomiasis, and LF [25–27]. Beyond the well-documented need to address prevailing rumors and generate demand through imparting knowledge about the infection, our results indicate that community engagement can benefit both reach and uptake. Interestingly, respondents discussed how a well-engaged community may have less need for additional "doorstep" education, saving time and reducing drug distributor workload as they have more time available to reach their allocated households. To our knowledge, considering community engagement as a time-saving approach that might reduce workload during drug distribution has not previously been discussed in the NTD literature. This presents a unique perspective potentially relevant to programs operating in more densely populated areas where drug administration represents a bigger portion of overall workload than travel or drug collection.

When discussing components of a successful community engagement strategy, we found that leveraging existing community structures such as chiefs and village elders to deliver messaging was vital, as it inferred legitimacy on the program and helped combat mistrust. This finding has been widely reported in previous studies that have highlighted the role of traditional and religious leaders in promoting trust [14,28,29]. In addition, and not previously discussed in the NTD literature, is the need for appropriate spacing for community engagement activities so as to give the messaging an opportunity to sufficiently circulate among community members. While the precise timing would depend on the local context, this finding further highlights the importance of forward planning for all components of an MDA program.

The negative impact that rumors and fears surrounding side-effects can have on uptake has been extensively reported in the literature and this study adds further evidence that many of these myths, such as fears of covert contraception or poisoning, are common across different program and country settings [14,21,29]. However, there has been less consideration of the drivers of these fears and how they can be addressed. Results from this study show that these fears are often linked to previous negative experiences of past MDAs and that community members are prone to conflate MDAs for other infections. This has implications for all MDA programs that are implementing in areas that have previously been targeted for MDA by other disease control programs. From these results, we recommend an increased awareness of the history of MDAs in target communities and the delivery of community engagement activities that directly and pre-emptively address any concerns that can be anticipated as arising from previous experiences from past MDAs. This approach is especially important within an elimination context, as more frequent rounds of MDA may be necessary to achieve successful outcomes.

## Limitations

This study has some limitations. Given the unique context of the TUMIKIA Project as a research study, some responses may have been specific to the conduct of the trial; however, we aimed to minimize the inclusion of data that was specific to trial implementation and focused our results on aspects of MDA delivery in order to maximize the relevance of our findings to other programs. Additionally, the data were collected in local languages; therefore, during the translation process, some dialect-specific nuances may have been lost. It is also possible that the three-month time period between MDA and data collection may have resulted in an increased occurrence of recall bias among respondents. Lastly, although the purpose of the trial was to understand community perspectives on deliver of community-wide MDA, the study did not conduct any interviews or FGDs in non-intervention clusters to understand their perception of the intervention.

## Conclusion

Findings from this study have shown there are a number of key strategies that community-wide MDA programs for STH should consider in order to achieve high reach and uptake. These results will be of most relevance for STH programs that are transitioning from control to elimination for which there is currently only limited literature examining what components are necessary for delivery of community-wide MDA. In the context of the wider literature surrounding MDA delivery and receipt, we have demonstrated that the challenges and solutions that are prevalent in other MDA-focused disease control programs such as schistosomiasis, LF and malaria are relevant to those faced in STH control. Our results suggest that while aspects of drug distributor recruitment benefit from professionalization, programs should not ignore the benefits that a ComDT approach can provide. A notable finding that adds to the literature is the role that previous community experiences of MDA of other NTDs can play in shaping trust in the treatment for a current STH program and the need to anticipate and preemptively address community fears resulting from these experiences. Taken all together, these findings suggest that a greater awareness of other PC control programs active in an area could be beneficial to program managers for all diseases targeted with PC as there is significant potential for cross-learning as well as a clear need to coordinate delivery.

## Supporting information

**S1 List. List of abbreviations.**
(DOCX)

## Acknowledgments

We would like to thank a number of key people for their valued input: Stefan Witek-McManus for input into development of the codebook and analysis plan; Janet Masaku and Rosemary Musuva for data coordination; and all the field officers, especially Hemed Mwakuzimui, Kennedy Kasbai, Naima Kamole and Dickson Muluu who conducted the transcription of the audio recordings. We appreciate the administrative and logistic support from Carlos Mcharo. We are very grateful to the Ministry of Health at the National level, particularly Sultani H Matendechero, Kefa Bota, Wycliffe Omondi, and Cecelia Wandera for oversight of the implementation of the MDA. We also gratefully acknowledge Athuman Chiguzo, Hajara El-Busaidy, and Redempta Muendo at from the Ministry of Health in Kwale for their leadership during the implementation as well as the tireless work from the health teams at the County, Sub-county, health facility and village level teams. Furthermore, the MDA would not have been

possible without the support of the Ministry of Interior and National Coordination of Government and the Public Service Office at the County, Sub-county, ward and village level. Finally, we thank the participants—CHAs, CHVs, and community members who took the time to provide valuable insights for this study.

## Author Contributions

**Conceptualization:** Katherine Halliday, Rachel Pullan, Doris Njomo.

**Data curation:** Hugo Legge.

**Formal analysis:** Hugo Legge, Stella Kepha, Mateo Prochazka, Marie-Claire Gwayi-Chore.

**Funding acquisition:** Rachel Pullan.

**Investigation:** Stella Kepha.

**Methodology:** Hugo Legge, Marie-Claire Gwayi-Chore, Doris Njomo.

**Project administration:** Hugo Legge, Stella Kepha.

**Supervision:** Katherine Halliday, Rachel Pullan, Marie-Claire Gwayi-Chore, Doris Njomo.

**Validation:** Marie-Claire Gwayi-Chore.

**Visualization:** Hugo Legge.

**Writing – original draft:** Hugo Legge, Stella Kepha, Mateo Prochazka, Marie-Claire Gwayi-Chore.

**Writing – review & editing:** Hugo Legge, Stella Kepha, Mateo Prochazka, Katherine Halliday, Rachel Pullan, Marie-Claire Gwayi-Chore, Doris Njomo.

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
