## [Decision Letter · Decision Letter 0]

31 Jan 2020

Dear Mr. Legge,

Thank you very much for submitting your manuscript "Implementer and recipient perspectives of community-wide mass drug administration for soil-transmitted helminths in Kwale County, Kenya" for consideration at PLOS Neglected Tropical Diseases. As with all papers reviewed by the journal, your manuscript was reviewed by members of the editorial board and by several independent reviewers. The reviewers appreciated the attention to an important topic. Based on the reviews, we are likely to accept this manuscript for publication, providing that you modify the manuscript according to the review recommendations. 

Sincerely,

Ana Lourdes Sanchez, PhD

Guest Editor

Hélène Carabin

Deputy Editor

Reviewer's Responses to Questions

**Key Review Criteria Required for Acceptance?**

**Methods**

-Are the objectives of the study clearly articulated with a clear testable hypothesis stated?

-Is the study design appropriate to address the stated objectives?

-Is the population clearly described and appropriate for the hypothesis being tested?

-Is the sample size sufficient to ensure adequate power to address the hypothesis being tested?

-Were correct statistical analysis used to support conclusions?

-Are there concerns about ethical or regulatory requirements being met?

Reviewer #1: This work is of qualitative analysis, my specialty in quantitative analysis limits methodological analysis.

Reviewer #2: Although the details are published elsewhere, I believe it would be relevant for readers to understand why the distribution occurred over an 8 day period, as this results in one of the major themes of the paper (increased workload by CHVs). 

One of the themes that surfaced was that community members were more receptive to CHVs that were identified as leaders, however the methods did not include how CHVs were selected (were they selected from the same community? were they of the same religion? spoke the same language? male/female? previous employment (nurses or doctors?)) - may be good to mention in a sentence or two). 

Other questions to consider or address: 

- Throughout the study, what was the retention of CHVs and were the CHVs delivering the drugs to the same communities each time; and what was the impact of this on uptake and reach?

- The study conducted interviews 3 months after the third round of MDA -- is there potential recall bias because of the long period of time in between intervention and interviews?

Reviewer #3: This paper reports a well laid out experimental study to see how a mass drug administration programme can move from a school based intervention to a community wide intervention. The study is designed to expand the intervention to entire communities which means there is a need to inform and organise communities on a much larger scale. It is not designed to submit to rigourous statistical analysis, but to reveal the issues that will arise for MDA in communities. The methods are clearly outlined and rationalised. Ethical approval is indicated.

**Results**

-Does the analysis presented match the analysis plan?

-Are the results clearly and completely presented?

-Are the figures (Tables, Images) of sufficient quality for clarity?

Reviewer #1: there is very little that can be highlighted from this type of study.

Reviewer #2: I really enjoyed the subheadings under both reach and uptake, I thought they were very creative and anecdotal. 

A few minor suggestions: 

pg 13-- "Let them be trained and be recognized": in your summary of this major theme, consider emphasizing the CHVs sense of pride, obligation and duty towards their community -- perhaps include these words in your discussion. This sense of purpose is something program managers can capitalize on during recruitment. 

Line 271: I think that this quotation emphasizes a theme not mentioned directly, but very much so implied: continuous feedback from CHAs to program implementers on training and other issues-- how would this open and two-way communication flow work, and what would it look like? Consider adding this to your list of suggestions/ table, as you have illustrated, it is vital for MDAs. 

Line 305: "drug distribution areas to be minimized" : could you clarify why they should be minimized? Wouldn't having more distribution areas (eg. one per community) make it easier for CHVs to deliver the drugs (instead of potentially having to transport them?) 

Discussion: Line 575: Absolutely love the clarity of the table, but found it rather redundant. Perhaps move the table into the results section, and eliminate some repetition from the discussion to make it more concise. 

- I understand the concept of keeping themes broad for generalizability, but I think it would have relevant and interesting if specific myths or educational misnomers/themes were included and how this compared to similar studies (perhaps educational campaigns can target specific myths eg. safety during pregnancy, how they work etc). 

- Overall well done.

Reviewer #3: The results are explained at length as questions asked by the participants and addressed in discussion. The take home message is really an understanding of the various message's that must be addresses when invoking community participation, particularly for mass drug administration. i.e. there is no diagnosis all are assumed infected and are subsequently treated. The training of personnel including volunteers (who incidentally are paid by the project). The difficulty of locating all residents and other important issues are mentioned.

**Conclusions**

-Are the conclusions supported by the data presented?

-Are the limitations of analysis clearly described?

-Do the authors discuss how these data can be helpful to advance our understanding of the topic under study?

-Is public health relevance addressed?

Reviewer #1: I emphasize that the work provides information on health policies related to STH but I do not consider it appropriate for this journal.

Reviewer #2: Great conclusion highlighting the importance of incorporating a community approach in large scale, potentially commercialized distribution.

Reviewer #3: The conclusions are somewhat depressing, as it appears almost impossible to effectively cover, find and treat whole communities. This is implicit in the conclusions. To this reviewer, the one omission is in the acknowledgements which omits the Kenya Government. If mass drug administration is to be applies (the paper talks of elimination), a major component will involve the Government, as there is clearly a need for national commitment and participation.

**Editorial and Data Presentation Modifications?**

Reviewer #1: (No Response)

Reviewer #2: Consider changing the word community "sensitization" to community education or engagement, as the sensitization process essentially involves a widespread educational campaign, and this wording has the potential to sound slightly condescending in global health settings.

Reviewer #3: My main criticism is that the authors use acronyms and abbreviation. If they insist on this there should be a glossary at the beginning

**Summary and General Comments**

Reviewer #1: None

Reviewer #2: I believe this article is extremely important for program implementers, logicians and others working in the field of global health.

Ethical Considerations: 

The information sheet for your study was provided in Swahili however interviews were conducted in many different languages. Further, was informed consent provided verbally due to potential illiteracy rates? Consider clarifying this information.

Overall, very thorough and well done.

Reviewer #3: It is a good study, and has several important conclusions

PLOS authors have the option to publish the peer review history of their article (what does this mean?). If published, this will include your full peer review and any attached files.

Reviewer #1: No

Reviewer #2: No

Reviewer #3: No
---

## [Editor Report · Decision Letter 1]

20 Mar 2020

Dear Mr. Legge,

Thank you very much for submitting your manuscript "Implementer and recipient perspectives of community-wide mass drug administration for soil-transmitted helminths in Kwale County, Kenya" for consideration at PLOS Neglected Tropical Diseases. As with all papers reviewed by the journal, your manuscript was reviewed by members of the editorial board and by several independent reviewers. The reviewers appreciated the attention to an important topic. Based on the reviews, we are likely to accept this manuscript for publication, providing that you modify the manuscript according to the review recommendations. 

We have reviewed the Revised version of this manuscripts and we are not ready to accept until we discuss the following reviewer points more thoroughly. There are no new reviews to respond to, but we ask that you respond further to these points:

REVIEWER #2: Consider changing the word community "sensitization" to community education or engagement, as the sensitization process essentially involves a widespread educational campaign, and this wording has the potential to sound slightly condescending in global health settings.

AUTHORS RESPONSE: We agree that there are many potential words that could be used for this process, however in this case we would prefer to retain the term as we feel that the term sensitization is a widely used and understood term in the NTD community when referring to MDA programmes.

EDITORS' OPINION: 

I agree with Reviewer 2 that the term sensitization is not the best choice for the activities that took place, which were basically information dissemination strategies to try and engage the community. Although the term has been used a few times in the NTD/STH literature, it is NOT widely used and understood. Actually, there is no agreed upon definition. A paper by Dierick et al (2018) defines it as “…community sensitizations, which is a process whereby research staff arrange meetings to make information on the research available in the villages from which potential research participants may be recruited” 

[Dierick et al. Community sensitization and decision‐making for trial participation: A mixed‐methods study from The Gambia. Developing World Bioeth. 2018;18:406–419. 10.1111/dewb.12160]

Even though the authors here elegantly refer to the engagement activities as “context-adapted sensitization strategy that leverages existing community structures and takes into consideration past community experiences of MDAs.” , in my opinion, community engagement would resonate better with the global health community.

On the other hand, sensitization was not seen by implementers as very well executed (“Implementers, for the most part, did not perceive sensitization activities to be effective in promoting community availability”). Similarly, sensitization was not enough to decrease mistrust among receivers: “While sensitization activities might adequately address perceived safety, efficacy and lack of knowledge, deep mistrust seemed to be a systemic problem”

I understand the reticence to accept this recommendation because the term sensitization is used more than 60 times in the manuscript. However, authors may want to keep in mind that “sensitization” is only the fourth “key actionable theme” emerging from data analysis, not the angular stone to MDA programs 

REVIEWER #2: Line 305: "drug distribution areas to be minimized" : could you clarify why they should be minimized? Wouldn't having more distribution areas (eg. one per community) make it easier for CHVs to deliver the drugs (instead of potentially having to transport them?) 

AUTHORS RESPONSE: We thank the reviewer for this comment but think there may have been a misunderstanding here. The sentence is in reference to a suggestion of a minimisation of the travel time between storage points and distribution areas. One method of achieving this is, as you suggest, to increase the storage sites where the CHAs are based with the drugs so as to reduce the distance between the drug collection point and villages where they are distributing. However, as this was not specifically referred to by the participant, we did not include this as a specific suggestion.

EDITORS' OPINION: 

The authors wrote: “CHVs emphasized the need for travel time between storage points and drug distribution areas be minimized”.

I can see why this wording can be confusing. To avoid it, I suggest rewording the sentence more simply to indicate that CHVs emphasized the need for reducing travel distance between storage points, which would result in maximizing resources and time utilization

Reviewer #3: My main criticism is that the authors use acronyms and abbreviation. If they insist on this there should be a glossary at the beginning

AUTHORS RESPONSE We agree that the manuscript contains a number of abbreviations and would be happy to provide a

list of abbreviations if there is a specified location where this section could be included according to the editors

EDITORS' OPINION: it would be appropriate to provide a list of abbreviations at the end of the manuscript.'

Sincerely,

Ana Lourdes Sanchez, PhD

Guest Editor

Hélène Carabin

Deputy Editor

We have reviewed the Revised version of this manuscripts and we are not ready to accept until we discuss the following reviewer points more thoroughly. There are no new reviews to respond to, but we ask that you respond further to these points:

REVIEWER #2: Consider changing the word community "sensitization" to community education or engagement, as the sensitization process essentially involves a widespread educational campaign, and this wording has the potential to sound slightly condescending in global health settings.

AUTHORS RESPONSE: We agree that there are many potential words that could be used for this process, however in this case we would prefer to retain the term as we feel that the term sensitization is a widely used and understood term in the NTD community when referring to MDA programmes.

EDITORS' OPINION: 

I agree with Reviewer 2 that the term sensitization is not the best choice for the activities that took place, which were basically information dissemination strategies to try and engage the community. Although the term has been used a few times in the NTD/STH literature, it is NOT widely used and understood. Actually, there is no agreed upon definition. A paper by Dierick et al (2018) defines it as “…community sensitizations, which is a process whereby research staff arrange meetings to make information on the research available in the villages from which potential research participants may be recruited” 

[Dierick et al. Community sensitization and decision‐making for trial participation: A mixed‐methods study from The Gambia. Developing World Bioeth. 2018;18:406–419. 10.1111/dewb.12160]

Even though the authors here elegantly refer to the engagement activities as “context-adapted sensitization strategy that leverages existing community structures and takes into consideration past community experiences of MDAs.” , in my opinion, community engagement would resonate better with the global health community.

On the other hand, sensitization was not seen by implementers as very well executed (“Implementers, for the most part, did not perceive sensitization activities to be effective in promoting community availability”). Similarly, sensitization was not enough to decrease mistrust among receivers: “While sensitization activities might adequately address perceived safety, efficacy and lack of knowledge, deep mistrust seemed to be a systemic problem”

I understand the reticence to accept this recommendation because the term sensitization is used more than 60 times in the manuscript. However, authors may want to keep in mind that “sensitization” is only the fourth “key actionable theme” emerging from data analysis, not the angular stone to MDA programs 

REVIEWER #2: Line 305: "drug distribution areas to be minimized" : could you clarify why they should be minimized? Wouldn't having more distribution areas (eg. one per community) make it easier for CHVs to deliver the drugs (instead of potentially having to transport them?) 

AUTHORS RESPONSE: We thank the reviewer for this comment but think there may have been a misunderstanding here. The sentence is in reference to a suggestion of a minimisation of the travel time between storage points and distribution areas. One method of achieving this is, as you suggest, to increase the storage sites where the CHAs are based with the drugs so as to reduce the distance between the drug collection point and villages where they are distributing. However, as this was not specifically referred to by the participant, we did not include this as a specific suggestion.

EDITORS' OPINION: 

The authors wrote: “CHVs emphasized the need for travel time between storage points and drug distribution areas be minimized”.

I can see why this wording can be confusing. To avoid it, I suggest rewording the sentence more simply to indicate that CHVs emphasized the need for reducing travel distance between storage points, which would result in maximizing resources and time utilization

Reviewer #3: My main criticism is that the authors use acronyms and abbreviation. If they insist on this there should be a glossary at the beginning

AUTHORS RESPONSE We agree that the manuscript contains a number of abbreviations and would be happy to provide a

list of abbreviations if there is a specified location where this section could be included according to the editors

EDITORS' OPINION: it would be appropriate to provide a list of abbreviations at the end of the manuscript.'
---

## [Editor Report · Decision Letter 2]

27 Mar 2020

Dear Mr. Legge,

We are pleased to inform you that your manuscript 'Implementer and recipient perspectives of community-wide mass drug administration for soil-transmitted helminths in Kwale County, Kenya' has been provisionally accepted for publication in PLOS Neglected Tropical Diseases.

Best regards,

Ana Lourdes Sanchez, PhD

Guest Editor

Hélène Carabin

Deputy Editor

---

## [Editor Report · Acceptance letter]

13 Apr 2020

Dear Mr. Legge,

We are delighted to inform you that your manuscript, "Implementer and recipient perspectives of community-wide mass drug administration for soil-transmitted helminths in Kwale County, Kenya," has been formally accepted for publication in PLOS Neglected Tropical Diseases.

Best regards,

Serap Aksoy

Editor-in-Chief

Shaden Kamhawi

Editor-in-Chief
